# Ethnic differences in the prevalence, socioeconomic and health related risk factors of knee pain and osteoarthritis symptoms in older Malaysians

**Sumaiyah Mat** [1], **Mohamad Hasif Jaafar**[1,2], **Chin Teck Ng** [3,4], **Sargunan Sockalingam**[5], **Jasmin Raja**[5], **Shahrul Bahyah Kamaruzzaman**[6], **Ai-Vyrn Chin**[6], **Azlina Amir Abbas**[7], **Chee Ken Chan**[7], **Noran Naqiah Hairi**[8], **Sajaratulnisah Othman**[9], **Robert G. Cumming**[10], **Nai Peng Tey**[11], **Maw Pin Tan**[1,6,12,13]*

1 Ageing and Age-Associated Disorders Research Group, Faculty of Medicine, University of Malaya, Kuala Lumpur, Malaysia, 2 Research Management Centre, Perdana University, Serdang, Selangor, Malaysia, 3 DUKE-NUS Medical School, Singapore, Singapore, 4 Singapore General Hospital, Singapore, Singapore, 5 Department of Rheumatology, Faculty of Medicine, University of Malaya, Kuala Lumpur, Malaysia, 6 Geriatric Division, Department of Medicine, Faculty of Medicine, University of Malaya, Kuala Lumpur, Malaysia, 7 National Orthopaedic Centre of Excellence in Research and Learning (NOCERAL), Department of Orthopaedic Surgery, Faculty of Medicine, University of Malaya, Kuala Lumpur, Malaysia, 8 Julius Centre University of Malaya, -Department of Social and Preventive Medicine, Faculty of Medicine, University of Malaya, Kuala Lumpur, Malaysia, 9 Department of Primary Care Medicine, Faculty of Medicine, University of Malaya, Kuala Lumpur, Malaysia, 10 School of Public Health, University of Sydney, Sydney, Australia, 11 Faculty of Economics and Administration, University of Malaya, Kuala Lumpur, Malaysia, 12 Centre for Innovation in Medical Engineering, University of Malaya, Kuala Lumpur, Malaysia, 13 Department of Medical Sciences, School of Healthcare and Medical Sciences, Sunway University, Bandar Sunway, Malaysia

* mptan@ummc.edu.my

**Data Availability Statement:** All Knee pain 1226 subjects Data files are available from the figshare database (accession number(s) https://figshare.

## Abstract

Knee pain is often underreported, underestimated and undertreated. This study was conducted to estimate the prevalence, burden and further identify socioeconomic factors influencing ethnic differences in knee pain and symptoms of OA among older adults aged 55 years and over in Greater Kuala Lumpur (the capital city of Malaysia). The sample for the Malaysian Elders Longitudinal Research (MELoR) was selected using stratified random sampling, by age and ethnicity from the electoral rolls of three parliamentary constituencies. Information on knee pain was available in 1226 participants, mean age (SD) 68.96 (1.57) years (409 Malay, 416 Chinese, 401 Indian). The crude and weighted prevalence of knee pain and self-reported knee OA symptoms were 33.3% and 30.8% respectively. There were significant ethnic differences in knee pain (crude prevalence: Malays 44.6%, Chinese 23.5% and Indians 31.9%, p<0.001). The presence of two or more non-communicable diseases (NCD) attenuated the increased risk of knee pain among the ethnic Indians compared to the ethnic Chinese. The prevalence of knee pain remained significantly higher among the ethnic Malays after adjustment for confounders. While the prevalence of knee pain in our older population appears similar to that reported in other published studies in Asia, the higher prevalence among the ethnic Malays has not previously been reported. Further research to determine potential genetic susceptibility to knee pain among the ethnic Malays is recommended.

com/articles/Knee_Pain_1226_subjects_sav/
9918257).

**Funding:** This work was funded by the
Fundamental Research Grants Scheme from the
Ministry of Education, Malaysia ((FRGS/1/2019/
SKK02/UM/01/1) and we would also like to
acknowledge the financial support provided by
University of Malaya under the Wellness Research
Centre (WRC) Impact-oriented Interdisciplinary
Research Grant (IIRG024-2019) and a High Impact
Research Grant from the Ministry of Education
(UM.C/625/1/HIR/MOHE/ASH/02) and University
of Malaya under the Wellness Research Centre
(WRC) grant challenge grant (GC002A-HTM). The
funders of this study played no part in the design,
data collection, data analysis or reporting of the
study.

**Competing interests:** The authors have declared
that no competing interests exist.

## Introduction

The population in low and middle-income countries is now ageing at a far quicker rate than in higher income countries which will represent 80% of older people population worldwide in 2050 [1]. This has been attributed to the dramatic decrease in birth rate and increase in life expectancy which has now exceeded 65 years in these countries [2]. The pattern of diseases affecting the global population is rapidly shifting from communicable to non-communicable diseases, with osteoarthritis (OA) being considered a non-communicable disease. Despite it being one of the most significant causes of disability world-wide [3], OA remains under-researched with limited information about its actual prevalence, and curative treatment is limited to joint replacement surgery. In fact, knee pain is often underreported, underestimated and undertreated [4].

The lack of a universal diagnostic criteria for OA has hampered research in this area [5]. Furthermore, OA can affect multiple joints and the affected joints may differ between individuals. The most commonly affected joints in OA are the knee and hip joints. The presence of joint pain provides the most convenient and easily measured surrogate for OA, particularly in population studies [6]. As osteoarthritis has been reported as the commonest cause of knee pain in older persons, such an approach would provide a reasonable estimate of the prevalence of knee OA [7]. However, knee pain may also arise from other causes including injury and inflammatory arthritis. Therefore, screening questions which include stiffness and function may improve the specificity of verbal screening questions.

The prevalence of knee pain has been reported as 25% to 56%, and appears to vary according to geographical location [8–10]. In a recent, larger survey, data from 3053 individuals in the Osteoarthritis Initiative (OAI) study reported the presence of at least mild knee pain in 67.1%. [11] African Americans are most likely to report knee pain. Serial surveys data from the National Health and Nutrition Examination Survey and Framingham Osteoarthritis study suggest that the prevalence of knee pain in the US has increased by 65% between 1983 and 2005, with non-Hispanic Blacks more likely to report knee pain [12]. In fact, a recent survey conducted among citizens aged 40 years and older in Germany found that three out of four respondents had experienced joint pain in the previous four weeks, of whom 30.9% reported knee pain [13]. Similarly, in Swedish adults age 56–84 years, the prevalence of frequent knee pain was 25.1% within a representative sample of 10,000 individuals.[9] In comparison, a study conducted in a rural area in China reported the prevalence of knee pain of 44% in men and 56% in women aged 50 years and above[10]. Available prevalence data from world-wide studies therefore suggest the possibility of ethnic variations in knee pain prevalence.

Previous community studies evaluating knee pain prevalence had been predominantly conducted within homogeneous populations, focussing on variations across age groups. In the present study, we estimated the overall prevalence of knee pain and OA symptoms among older community dwellers within a multi-ethnic middle-income South East Asian nation. In addition, we evaluated the role of socioeconomic factors in determining the ethnic differences in knee pain and OA symptoms within this population.

## Methods

### Study population

Data for this analysis was taken from a cross-sectional survey of the first wave of the Malaysian Elders Longitudinal Research (MELoR) study. The MELoR study was designed as a longitudinal cohort study based in Kuala Lumpur and its satellite town, Petaling Jaya, in the Klang Valley [14]. Individuals aged 55 years and above were selected through stratified random

sampling according to the three main ethnic groups and by age deciles. The electoral rolls of the Parliamentary constituencies of Petaling Jaya North, Petaling Jaya South, and Lembah Pantai were used as the sampling frame. Data was collected between November 2013 to October 2015. This study was approved by University of Malaya Medical Centre Medical Ethics Committee (MEC 943.6) and complied with the Helsinki Declaration of 1975, revised in 1983. Written informed consent was obtained from all study participants prior to their inclusion. The inclusion criteria were age 55 years and above, able to provide informed consent, and belonging to one of the three major ethnic groups of Malay, Chinese or Indian. Institutionalized older adults and those with communication difficulties, including cognitive impairment, affecting their ability to respond to the questionnaire, were excluded.

## Data collection

Participants were contacted and visited at their own homes initially to recruit them into the study. A structured interview using a computer aided questionnaire was completed during the visit. Participants were then requested to visit the hospital for a detailed health check.

## Case definitions

The presence of knee pain was established when participants were asked to indicate whether they had any pain in the head, back, hip, knees, feet, mouth/teeth or all over. Severity of pain was then determined by asking participants whether the pain specific to the affected body part or parts was "mild", "moderate" or "severe". Those with knee pain were asked to rate pain severity separately for the left and right.

In order to determine the presence of OA symptoms, participants were also asked the question, "in the past 12 months, have you had pain, aching, or stiffness in either knee, on most days for at least one month?" Participants were required to report pain, aching and stiffness in or around the knee, including the front, back or side of the knee. This question yielded a dichotomous "Yes" or "No" response.

## Socioeconomic and health-related variables

A computer-based questionnaire was administered during the home-based interview. Information about age, sex, ethnicity, marital status, educational level, type of occupation, and self-reported history of non-communicable disorders, including hypertension, diabetes mellitus (DM), stroke, heart disease, hyperlipidemia, asthma, bronchitis, chronic obstructive pulmonary disease (COPD), cancer and Parkinson's disease were obtained during the home visit.

Body weight, height and physical performance measurements were obtained during the hospital health check. The body mass index (BMI) was calculated by dividing the body weight in kilograms with square of the height in metres. Obesity was defined as a body mass index (BMI) of $30kg/m^2$ or above.

## Statistical analysis

Data analyses were conducted using SPSS Version 20 (IBM, Armonk, NY, USA). The sample population were divided into five years age bands: 55–59, 60–64, 65–69, 70–74, and ≥75 years. The chi-square test was utilized to test for significant associations between age groups and presence of knee pain and OA symptoms. It was necessary add sample weights to take into account the disproportionate sampling for the ethnic groups and age groups. Sample weights for five-year age-groups were calculated according to the population Kuala Lumpur from the 2010 census [15]. We report crude and age-standardized prevalence for the overall population

and separately for each ethnicity. Prevalence for individual five-year age groups were also provided. Logistic regression analyses were then employed to adjust for potential confounders. To explore the potential rationale underlying ethnic differences, the relative change in the odds ratio (OR) between the base model and subsequently models was determined using the formula: $[(OR_u-OR_a)/(OR_u)]$, where $OR_u$ is the OR for the unadjusted odds ratio for knee pain and $OR_a$ is the OR after adjustment for the subsequent variable for all three ethnic groups.

## Results

### Prevalence of knee pain and symptoms of knee osteoarthritis

Information on knee pain was available for 1226 participants. Of the 1226 participants, 408 (33.3%) reported the presence of knee pain. Three hundred and thirty-two (27%) reported the presence of symptoms of knee osteoarthritis. The crude prevalence of knee pain for the overall study population and age categories are summarized in Fig 1. The overall estimated weighted prevalence for individuals aged 55 years adjusted to the population in Kuala Lumpur was 30.8% for knee pain and 25.4% for pain and stiffness in and around the knee lasting at least a month over the past 12 months. (Fig 1)

### Prevalence of knee pain and knee osteoarthritis by ethnicity

There were significant differences in proportion individuals with knee pain (p<0.001) and knee arthritis symptoms (p<0.001) between ethnic groups. The weighted prevalence of knee pain among the ethnic Malays was 45.6%, ethnic Indians 31.5% and ethnic Chinese 21.7%. The weighted prevalence of knee OA symptoms among the ethnic Malays was 37.7%, ethnic Indians 25.7% and ethnic Chinese 17.9%. (Fig 2)

### Univariate analysis for factors associated with knee pain and knee osteoarthritis

Table 1 shows the basic characteristics of the study population and the univariate odd ratios for sociodemographic factors influencing knee pain and knee OA symptoms. There were significant sex and ethnic differences in knee pain and knee OA symptoms. Being female and

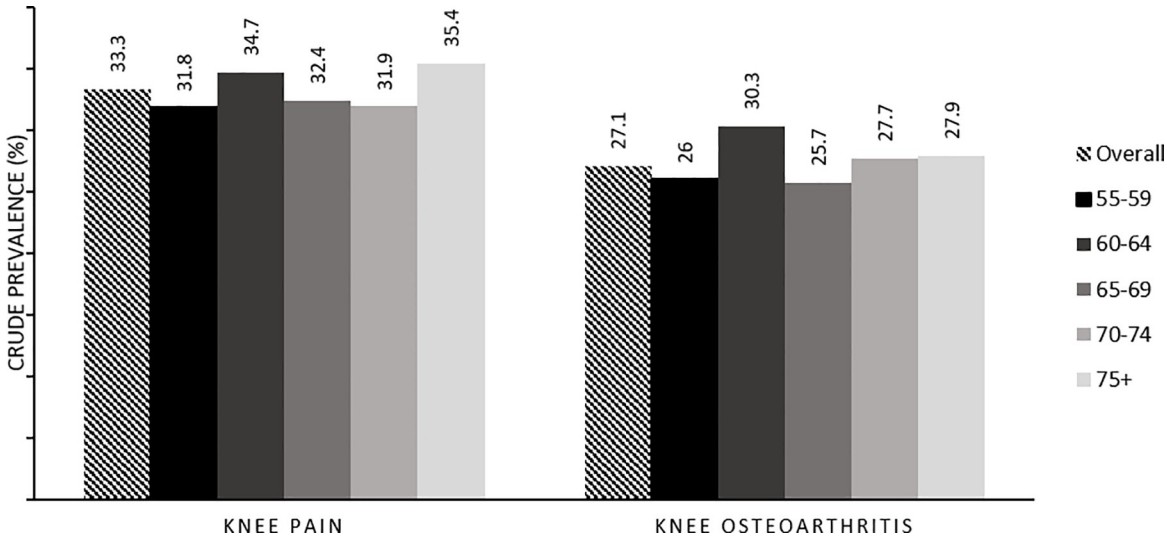

**Fig 1. Crude prevalence of knee pain and knee osteoarthritis by age.**

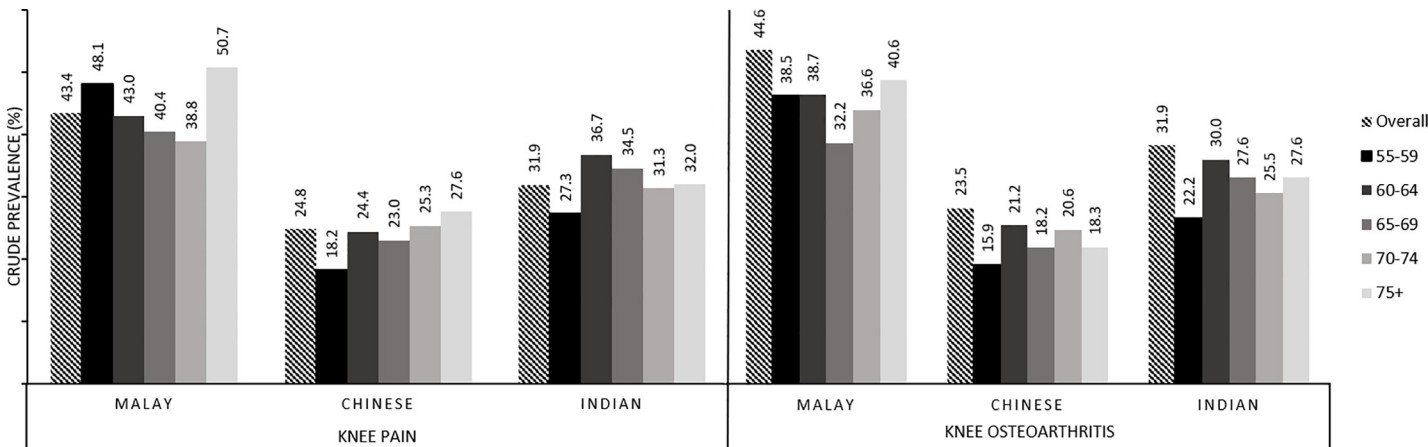

**Fig 2. Overall crude prevalence of knee pain and knee osteoarthritis by ethnicity according to age group.**

belonging to the Malay or Indian ethnic groups were associated with increased risk of knee pain. Having no partner and lower educational levels were also significantly associated with knee pain. In evaluating the type of occupation as risk factors, compared to professionals, participants who were not working, service and sales workers, plant and machine operators were more likely to report knee pain. The presence of two or more non-communicable disorder (NCD), obesity or polypharmacy were significantly associated with increased risk of knee pain and knee OA symptoms. (Table 1) Heart disease, cerebrovascular disease, peripheral vascular disease, hypertension, diabetes, hyperlipidemia were the NCD measured in this study. Polypharmacy was defined as the consumptions of five or more medications.

### Multivariate analyses for ethnic differences

Further analyses allowed us to evaluate the sociodemographic influences on ethnic disparities in knee pain and OA symptoms. This analysis was conducted using logistic regression with dummy variables using Chinese ethnicity as the reference group. In the unadjusted model, the ethnic Malays were significantly more likely to report knee pain in unadjusted analysis and this relationship remained significant after adjustment for sociodemographic differences, two or more NCD and obesity, compared to the ethnic Chinese. The increased likelihood of knee pain observed among the ethnic Indians, however, did not withstand adjustment for the presence of two or more NCD, suggesting that the increase presence of knee pain in the ethnic Indians compared to the ethnic Chinese was attributable to the increased risk of NCD among the ethnic Indians. The presence of two or more NCD in fact accounted for 8.97% of the ethnic disparities in knee pain between Indians and the Chinese, and 4.62% of the ethnic disparities in knee pain between Malays and Chinese.

As for the presence of knee OA symptoms, the increased association in the Malay ethnicity compared to the ethnic Chinese remained independent after adjustment for age, sex, marital status, education, two or more non-communication disorders and obesity. The increased association in knee OA symptoms among the ethnic Indians compared to the ethnic Chinese did not withstand adjustment for two or more NCD, with the presence of two or more NCD accounting for 9.10% of the disparities in knee OA symptoms between the ethnic Indians and ethnic Chinese, after taking into account age, sex, marital status and educational level. (Table 2)

**Table 1. Univariate analysis for factors associated with knee pain and knee osteoarthritis.**

| | N (%) | Knee Pain (n = 1226) | Knee OA (n = 1204) |
|---|---|---|---|
| Ethnicity | | | |
| Chinese | 416 (33.9) | 1 | 1 |
| Malay | 409 (33.4) | **2.38 (1.77–3.02)** | **2.48 (1.80–3.41)** |
| Indian | 401 (32.7) | **1.50 (1.10–2.03)** | **1.54 (1.10–2.14)** |
| Age (years), Mean (SD) | 68.97 (7.48) | 1.00 (0.99–1.02) | 1.00 (0.98–1.02) |
| Female | 694 (56.6) | **1.82 (1.42–2.33)** | **1.95 (1.49–2.54)** |
| Marital status | | | |
| Single | 337 (27.5) | **1.38 (1.07–1.80)** | 1.29 (0.98–1.70) |
| Spouse/Partner | 887 (72.3) | 1 | 1 |
| Education level | | | |
| Primary and lower | 342 (27.9) | **1.86 (1.44–2.41)** | **1.82 (1.39–2.38)** |
| Secondary and above | 883 (72.0) | 1 | 1 |
| Occupation (past/present) | | | |
| Professional | 317 (25.9) | 1 | 1 |
| Not working/ Housewife | 121 (9.9) | **2.23 (1.44–3.46)** | **1.80 (1.13–2.87)** |
| Executive officer | 71 (5.8) | 1.20 (0.68–2.13) | 1.01 (0.55–188) |
| Technician | 70 (5.7) | 0.92 (0.50–1.68) | 1.03 (0.56–1.92) |
| Craft and related trade worker | 31 (2.5) | 1.81 (0.84–3.89) | 2.16 (0.99–4.67) |
| Elementary occupation | 79 (6.4) | 1.41 (0.83–2.39) | 0.97 (0.53–1.77) |
| Manager/administrator | 156 (12.7) | 1.20 (0.78–1.84) | 1.01 (0.64–1.60) |
| Clerk | 104 (8.5) | 1.45 (0.90–2.35) | 1.23 (0.74–2.06) |
| Service and sales worker | 155 (12.6) | **1.67 (1.10–2.52)** | **1.69 (1.10–2.60)** |
| Plant and machine operator/driver | 116 (9.5) | **2.58 (1.66–4.02)** | **2.28 (1.44–3.61)** |
| Comorbidities | | | |
| Hypertension | 654 (53.3) | **1.65 (1.29–2.10)** | **1.72 (1.33–2.22)** |
| Diabetes | 382 (31.2) | **1.57 (1.22–2.02)** | **1.48 (1.13–1.93)** |
| Hyperlipidemia | 672 (54.8) | **1.71 (1.34–2.18)** | **1.57 (1.21–2.03)** |
| Ischemic Heart Disease | 148 (12.1) | 0.96 (0.66–1.38) | 0.97 (0.66–1.44) |
| Asthma | 96 (7.8) | 1.11 (0.72–1.72) | 1.25 (0.80–1.97) |
| Bronchitis | 26 (2.1) | 1.26 (0.57–2.80) | 1.49 (0.65–3.41) |
| Chronic Obstructive Pulmonary Disease (COPD) | 10 (0.8) | 1.34 (0.38–4.78) | 1.76 (0.49–6.28) |
| Parkinson | 6 (0.5) | 0.40 (0.05–3.43) | 0.52 (0.06–4.50) |
| Stroke | 21 (1.7) | 1.24 (0.51–3.01) | 1.32 (0.53–3.30) |
| Cancer | 67 (5.5) | 0.78 (0.45–1.35) | 0.93 (0.53–1.64) |
| ≥2 NCD | 919 (75.0) | **2.51 (1.84–3.43)** | **2.84 (2.00–4.04)** |
| Obesity (BMI>30) | 152 (12.4) | **2.57 (1.81–3.64)** | **2.53 (1.77–3.62)** |
| ≥5 medications | 436 (35.6) | **1.72 (1.34–2.21)** | **1.83 (1.40–2.38)** |

OA = osteoarthritis symptoms; NCD = non-communicable disease; BMI = body mass index

## Discussions

This was the first study in Asia to report direct ethnic comparisons in knee pain and knee OA symptoms within the same geographical area. Knee pain was present in 30.8% the population aged 55 years and over residing in greater Kuala Lumpur, while symptoms of pain and stiffness in and around the knee lasting at least a month over the preceding 12 months were reported

**Table 2. Multivariate analysis for ethnicity differences in knee pain.**

|  | unadjusted | Adjustment 1 |  | Adjustment 2 |  | Adjustment 3 |  | Adjustment 4 |  |
| --- | --- | --- | --- | --- | --- | --- | --- | --- | --- |
| **Knee Pain** | OR (95% CI) | OR (95% CI) | % | OR (95% CI) | % | OR (95% CI) | % | OR (95% CI) | % |
| Chinese | 1 | 1 |  | 1 |  | 1 |  | 1 |  |
| Malay | **2.38 (1.77–3.02)** | **2.36 (1.72–3.20)** | 0.84 | **2.36 (1.71–3.24)** | 0.84 | **2.27 (1.66–3.12)** | 4.62 | **2.21 (1.55–3.14)** | 7.14 |
| Indian | **1.50 (1.10–2.03)** | **1.54 (1.12–2.11)** | -2.67 | **1.53 (1.11–2.12)** | -2.00 | 1.36 (0.99–1.88) | 8.97 | 1.29 (0.91–1.82) | 14.0 |
| Knee OA |  |  |  |  |  |  |  |  |  |
| Chinese | 1 | 1 |  | 1 |  | 1 |  | 1 |  |
| Malay | **2.48 (1.80–3.41)** | **2.47 (1.76–3.45)** | 0.40 | **2.61 (1.84–3.67)** | -5.24 | **2.39 (1.70–3.35)** | 3.63 | **2.26 (1.55–3.29)** | 8.87 |
| Indian | **1.54 (1.10–2.14)** | **1.60 (1.14–2.25)** | -3.90 | **1.66 (1.17–2.37)** | -7.79 | 1.40 (0.99–1.99) | 9.10 | 1.28 (0.88–1.87) | 16.88 |

OR = odds ratio; CI = confidence interval, OA = Osteoarthritis

% = proportion of disparities explained [(OR $_{unadjusted}$- OR $_{adjusted}$) /(OR $_{unadjusted}$)] *100%

Adjustment 1: adjusted for age, sex, marital status, education

Adjustment 2: adjusted for age, sex, marital status, education, occupation

Adjustment 3: adjusted for age, sex, marital status, education,≥2 non-communicable disorders

Adjustment 4: adjusted for age, sex, marital status, education, ≥2 non-communicable disorders, and obesity

by 25.4% of the study population. Large ethnic differences were observed with 43.4% of the ethnic Malays reporting knee pain, compared to 31.9% of the ethnic Indians and 24.8% of the ethnic Chinese populations. The corresponding prevalence of OA symptoms was also higher among the ethnic Malays at 44.6%, compared to the ethnic Indians at 31.9% and ethnic Chinese at 25.7%.

Twenty-seven percent of the older UK and 22.7% US population [16] reported the presence of knee pain. The prevalence of knee pain among older Asian populations for China was in the range of 11% to 56% [10, 17, 18], Japan 33% [19], Korea 38% [20], and Vietnam 61% [21]. Direct comparisons between the studies conducted in different continents cannot be made due to differences in study methodology and sample populations. Recent meta-analysis on total of 28 studies showed that differences exist in clinical pain severity between African American and Non-hispanic White with OA.[22] The increased propensity to developing OA of the knee among the Chinese population compared to Caucasians had also previously been reported [23]. These differences could be attributable to culture, pain threshold, genetic predisposition [20] and environmental or lifestyle risk factors [24]. A cohort study conducted among older people aged 60 years in Beijing found that prolonged squatting for more than an hour per day when they were 25 years of age was common, and people who reported squatting more than 3 hours per day had twice the risk of developing knee OA compared to those who reported squatting less than 30 minutes a day [25].

The high prevalence of knee OA among the ethnic Malays compared to their ethnic Chinese and Indian counterparts has not previously been reported [23]. In addition, the prevalence of knee pain in our ethnic Chinese population appeared lower than that reported in mainland China. The rationale behind these ethnic differences are not apparent, but may be explained by lifestyle and occupation. Adjustment for occupational differences between the difference ethnic groups, however, did not influence the ethnic differences in knee symptoms, suggesting that occupational differences had a limited role in the ethnic differences in knee joint degeneration in our older population.

Like Asian populations, the ethnic Malays practice the 'floor culture' [26]. Those who frequently sit on the floor crossed-legged or on bended knees tend to be at higher risk of knee injuries [27, 28]. Recurrent knee trauma may then increase the risk of OA at older age [29].

Since it is common among Asians, including Malays to practice the floor culture within their lifestyle [30], it has been suggested that 'floor culture' might be a risk factor for knee pain. It was not possible to objectively measure 'floor culture' practices in our study sample. However, the excess risk for knee pain and knee OA symptoms in our ethnic Malays compared to the ethnic Chinese have remained unexplained despite adjustments for socioeconomic factors, obesity and NCD. It, therefore, remains plausible that the ethnic Malays are at an increased risk of OA due to lifestyle factors including the "floor culture", though, genetic differences may also explain these differences.

The increased risk of knee pain and knee OA symptoms among the ethnic Indians in our study was accounted for by the presence of two or more NCD. Previous longitudinal studies have suggested the presence of an association between NCD, such as DM, with the presence and progression of OA [31–33]. Most of the studies included in a recent systematic review had reported a significant relationship between DM and OA [34]. Ethnic Indians living in Southeast Asia are known to be at an increased risk of DM compared to the other ethnic groups, which is consistent with the findings of our study [35, 36].

The prevalence of knee pain did not increase with increasing age. A previous study in conducted in the US suggested that chronic pain stopped increasing once individuals hit 60 years of age [37]. The lack of increase or apparent reduction in knee OA with age may suggest a survival advantage among those without OA, hence despite new cases of knee OA emerging with increasing age, there is no overall increase in the prevalence of OA among the older old compared to the younger old [38]. With rapid development, the traditional agricultural lifestyle is disappearing and the population is now mainly working in sedentary jobs in the manufacturing and service sectors. In line with this, the burden of NCD is now increasing. Recent studies have linked OA with metabolic syndrome. Therefore, the rise in knee OA among the young old ethnic Malays may reflect the corresponding alarming rise in obesity, DM and metabolic disorders in this group [36], though our multivariate analysis suggests that the overall increased susceptibility to knee pain and knee OA among the ethnic Malays has not been fully explained by the factors adjusted for, including NCD and obesity.

As the knee pain and knee OA questions were embedded within the extensive questionnaire of the first wave of a cohort study, the questions were administered by trained enumerators. The diagnosis of OA could not, therefore, be confirmed by a medical expert. It was also logistically not possible to perform knee radiographs on all our participants, but it is now considered well-established that radiographic changes of OA correlate poorly with symptomatic presentations [39]. Furthermore, as the participants were sampled from an urban population, our findings may not be generalizable to rural populations. Our findings were also based on self-reported symptoms which may lead to ethnic biases in response. Future research should seek to now unravel the rationale behind the differences in ethnic prevalence of knee pain and OA symptoms in our population.

The estimated prevalence of knee pain among individuals aged 55 years and older residing in Kuala Lumpur, Malaysia was 30.8%, while the prevalence of self-reported OA symptoms over the past year was 25.4%. The prevalence of knee pain was highest among the ethnic Malays with an estimated prevalence of 45.6%. In general, knee pain and OA related symptoms were associated with female gender, educational level, occupation, comorbidities and obesity. The presence of two or more NCD accounted for the increased risk of knee pain among the ethnic Indians compared to the ethnic Chinese. The Malay ethnicity was independently associated with knee pain and knee OA symptoms. Further research should evaluate the potential genetic and non-genetic mechanisms underlying the increased likelihood of knee pain and knee OA symptoms among the ethnic Malays.

## Author Contributions

**Conceptualization:** Sumaiyah Mat, Sargunan Sockalingam, Shahrul Bahyah Kamaruzzaman, Ai-Vyrn Chin, Azlina Amir Abbas, Noran Naqiah Hairi, Sajaratulnisah Othman, Robert G. Cumming, Nai Peng Tey, Maw Pin Tan.

**Formal analysis:** Sumaiyah Mat, Mohamad Hasif Jaafar, Chin Teck Ng, Sajaratulnisah Othman, Robert G. Cumming, Nai Peng Tey.

**Funding acquisition:** Shahrul Bahyah Kamaruzzaman, Maw Pin Tan.

**Investigation:** Sumaiyah Mat, Noran Naqiah Hairi, Maw Pin Tan.

**Methodology:** Sargunan Sockalingam, Ai-Vyrn Chin, Noran Naqiah Hairi, Robert G. Cumming, Nai Peng Tey, Maw Pin Tan.

**Supervision:** Maw Pin Tan.

**Writing – original draft:** Sumaiyah Mat, Mohamad Hasif Jaafar, Chin Teck Ng, Sargunan Sockalingam, Jasmin Raja, Shahrul Bahyah Kamaruzzaman, Ai-Vyrn Chin, Azlina Amir Abbas, Chee Ken Chan, Noran Naqiah Hairi, Sajaratulnisah Othman, Robert G. Cumming, Nai Peng Tey, Maw Pin Tan.

**Writing – review & editing:** Sumaiyah Mat, Mohamad Hasif Jaafar, Chin Teck Ng, Shahrul Bahyah Kamaruzzaman, Azlina Amir Abbas, Chee Ken Chan, Noran Naqiah Hairi, Sajaratulnisah Othman, Maw Pin Tan.

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
