## [Decision Letter · Decision Letter 0]

23 Aug 2019

PONE-D-19-15422

Ethnic differences in the Prevalence, Socioeconomic and Health Related Risk Factors of Knee Pain and Osteoarthritis Symptoms in Older Malaysians

PLOS ONE

Dear Dr. Tan,

Thank you for submitting your manuscript to PLOS ONE. After careful consideration, we feel that it has merit but does not fully meet PLOS ONE’s publication criteria as it currently stands. Therefore, we invite you to submit a revised version of the manuscript that addresses the points raised during the review process.

We would appreciate receiving your revised manuscript by Oct 07 2019 11:59PM. To enhance the reproducibility of your results, we recommend that if applicable you deposit your laboratory protocols in protocols.io, where a protocol can be assigned its own identifier (DOI) such that it can be cited independently in the future. For instructions see: http://journals.plos.org/plosone/s/submission-guidelines#loc-laboratory-protocols

We look forward to receiving your revised manuscript.

Kind regards,

Yan Li

Academic Editor

PLOS ONE

Journal Requirements:

2. Thank you for your ethics statement :

"This study was approved by Medical Ethics Committee (MEC 943.6) and complied with the Helsinki Declaration of 1975, revised in 1983."

Once you have amended this statement in the Methods section of the manuscript, please add the same text to the “Ethics Statement” field of the submission form (via “Edit Submission”).

This work was funded by the High Impact Research Grant from the Ministry of Education, Malaysia ((UM.C/625/1/HIR/MOHE/ASH/02) and we would also like to acknowledge the financial support provided by University of Malaya under the Wellness Research Centre (WRC) grant challenge grant (GC002A-HTM).The funders of this study played no part in the design, data collection, data analysis or reporting of the study. They supplied funding for the salaries of SM and MHJ and for the purchase of equipment, consumables and transportation required for the conduct of the study.

"NO - Include this sentence at the end of your statement: The funders had no role in

study design, data collection and analysis, decision to publish, or preparation of the

manuscript".

Reviewers' comments:

Reviewer's Responses to Questions

**Comments to the Author**

1. Is the manuscript technically sound, and do the data support the conclusions?

Reviewer #1: Yes

Reviewer #2: Partly

2. Has the statistical analysis been performed appropriately and rigorously? 

Reviewer #1: Yes

Reviewer #2: Yes

3. Have the authors made all data underlying the findings in their manuscript fully available?

Reviewer #1: Yes

Reviewer #2: Yes

4. Is the manuscript presented in an intelligible fashion and written in standard English?

Reviewer #1: Yes

Reviewer #2: Yes

5. Review Comments to the Author

Reviewer #1: It's an interesting study which revealed the common symptoms in an epidemiological way. It's well written and technically sound.

Reviewer #2: The introduction and discussion parts need to be re-written, most of the references are really old, please up to date,

6. PLOS authors have the option to publish the peer review history of their article (what does this mean?). If published, this will include your full peer review and any attached files.

Reviewer #1: No

Reviewer #2: No

---

## [Author Response · Author response to Decision Letter 0]

7 Oct 2019

Reviewer 1 

It's an interesting study which revealed the common symptoms in an epidemiological way. It's well written and technically sound. 

Response: Thank you.

Reviewer 2 

The introduction and discussion parts need to be re-written, most of the references are really old, please up to date,

Response: 

Thank you for your suggestions, I have updated with latest references as below: 

United Nations DoEaSA, Population Division,. World Population Prospects: The 2019 revision 2019. Available from: https://population.un.org/wpp/Graphs/Probabilistic/.

Page 3 Introduction part, paragraph 1 line 3, 

Thomas E, Peat G, Croft P. Defining and mapping the person with osteoarthritis for population studies and public health. Rheumatology (Oxford). 2014;53(2):338-45. Epub 2013/10/29. doi: 10.1093/rheumatology/ket346. PubMed PMID: 24173433

Page 3 Introduction part, Paragraph 2, line 14

Barbour KE, Helmick CG, Boring M, Brady TJ. Vital Signs: Prevalence of Doctor-Diagnosed Arthritis and Arthritis-Attributable Activity Limitation - United States, 2013-2015. MMWR Morb Mortal Wkly Rep. 2017;66(9):246-53. Epub 2017/03/10. doi: 10.15585/mmwr.mm6609e1. PubMed PMID: 28278145; PubMed Central PMCID: PMCPMC5687192

Page 4 Introduction part, Paragraph 2, line 14

Vaughn IA, Terry EL, Bartley EJ, Schaefer N, Fillingim RB. Racial-Ethnic Differences in Osteoarthritis Pain and Disability: A Meta-Analysis. J Pain. 2019;20(6):629-44. Epub 2018/12/14. doi: 10.1016/j.jpain.2018.11.012. PubMed PMID: 30543951; PubMed Central PMCID: PMCPMC6557704.

Page 14 Discussions part, Paragraph 2, line 16

Liu Q, Wang S, Lin J, Zhang Y. The burden for knee osteoarthritis among Chinese elderly: estimates from a nationally representative study. Osteoarthritis Cartilage. 2018;26(12):1636-42. Epub 2018/08/22. doi: 10.1016/j.joca.2018.07.019. PubMed PMID: 30130589 

Page 14, Discussion part, Paragraph 2, line 17

---

## [Decision Letter · Decision Letter 1]

29 Oct 2019

Ethnic differences in the Prevalence, Socioeconomic and Health Related Risk Factors of Knee Pain and Osteoarthritis Symptoms in Older Malaysians

PONE-D-19-15422R1

Dear Dr. Tan,

We are pleased to inform you that your manuscript has been judged scientifically suitable for publication and will be formally accepted for publication once it complies with all outstanding technical requirements.

With kind regards,

Yan Li

Academic Editor

PLOS ONE

Additional Editor Comments (optional):

Reviewers' comments:

Reviewer's Responses to Questions

**Comments to the Author**

1. If the authors have adequately addressed your comments raised in a previous round of review and you feel that this manuscript is now acceptable for publication, you may indicate that here to bypass the “Comments to the Author” section, enter your conflict of interest statement in the “Confidential to Editor” section, and submit your "Accept" recommendation.

Reviewer #1: All comments have been addressed

Reviewer #2: All comments have been addressed

2. Is the manuscript technically sound, and do the data support the conclusions?

Reviewer #1: Yes

Reviewer #2: Yes

3. Has the statistical analysis been performed appropriately and rigorously? 

Reviewer #1: Yes

Reviewer #2: Yes

4. Have the authors made all data underlying the findings in their manuscript fully available?

Reviewer #1: Yes

Reviewer #2: Yes

5. Is the manuscript presented in an intelligible fashion and written in standard English?

Reviewer #1: Yes

Reviewer #2: Yes

6. Review Comments to the Author

Reviewer #1: All concerns were well addressed. The introduction and discussion parts are re-written and updated. No further questions. Agree to publish.

Reviewer #2: the authors have address the previous comments, I don't have any other comments, recommend to be accepted

7. PLOS authors have the option to publish the peer review history of their article (what does this mean?). If published, this will include your full peer review and any attached files.

Reviewer #1: No

Reviewer #2: No

---

## [Editor Report · Acceptance letter]

12 Nov 2019

PONE-D-19-15422R1 

Ethnic differences in the Prevalence, Socioeconomic and Health Related Risk Factors of Knee Pain and Osteoarthritis Symptoms in Older Malaysians 

Dear Dr. Tan:

I am pleased to inform you that your manuscript has been deemed suitable for publication in PLOS ONE. Congratulations! Your manuscript is now with our production department. 

With kind regards,

on behalf of

Dr. Yan Li 

Academic Editor

PLOS ONE